# Study of Material Properties and Creep Behavior of a Large Block of AISI 316L Steel Produced by SLM Technology

**Šárka Hermanová [1,\*], Zdeněk Kuboň [1] , Petr Čížek [1], Jana Kosňovská [1], Gabriela Rožnovská [1], Ondřej Dorazil [1] and Marcela Cieslarová [2]**

1  Material and Metallurgical Research Ltd., Pohraniční 693/31, 703 00 Ostrava, Czech Republic; creep.lab@mmvyzkum.cz (Z.K.); petr.cizek@mmvyzkum.cz (P.Č.); jana.kosnovska@mmvyzkum.cz (J.K.); gabriela.roznovska@mmvyzkum.cz (G.R.); ondrej.dorazil@mmvyzkum.cz (O.D.)

2  Strojírny a stavby Třinec, a.s., 739 61 Třinec, Czech Republic; marcela.cieslarova@trz.cz

\*  Correspondence: sarka.hermanova@mmvyzkum.cz; Tel.: +42-0595-952-068

**Abstract:** The additive manufacturing (3D printing) of metallic materials is a relatively new technology and its use is quickly increasing. Although it is of interest to many researchers, there are still areas which are not fully explored. One of those areas is the behavior of large components and/or semi-products processed by 3D printing. This work is focused on the study of material properties of additive manufactured large block made of AISI 316L steel in two heat treatment conditions (as-printed and solution annealed) and their comparison with the properties of hot-rolled plate performed by tensile tests, Charpy V-notch tests, small punch tests and stress rupture tests. Mechanical tests were complemented by microstructural investigation and the fractographic analysis of fracture surfaces. We found out that mechanical and long-term properties of large 3D printed blocks of this steel are excellent and comparable with other published results obtained on small-sized and intentionally produced test pieces. The observed lower ductility is the result of printing imperfections in microstructure. The results of small punch tests confirmed the possibility of exploiting the existing database and using the correlation between small punch tests and tensile tests results even for 3D-printed AISI 316L steel.

**Keywords:** additive manufacturing; steel AISI 316L; mechanical properties; small punch tests; stress-rupture tests

## 1. Introduction

Additive manufacturing, widely known also as 3D printing, is relatively new technology. It is a process of creating a three-dimensional object layer by layer from a digital file (CAD models) that was at first applied to plastic material and for producing models and prototypes. The history of the 3D printing process, which dates back to 1980s, is described, for instance, in [1–3].

The 3D printing of metallic materials is one of the promising ways of increasing the competitiveness of companies in the engineering industry, especially in the production of shape-complicated parts for material savings and the reduction of the material and processing costs. However, before companies include these innovative technologies in their product portfolio, designers, in particular, must know the properties of the parts produced this way and guarantee their conformity with standards. Regarding the future use, the important material properties can also include long-term creep properties.

There are many types of additive manufacturing processes. One of them is metal powder bed fusion technology, as the International Committee ASTM F42 classified, and it is included in the ISO/ASTM 52900:2015 [4] standard. Selective laser melting (SLM) as a part of the powder bed fusion was used to produce our experimental material.

Selective laser melting is a technique designed to use a high-power density laser to melt and fuse metallic powders together [5]. This technology can be used for special

production in aerospace or automotive components, especially in cases where it is difficult to manufacture them by conventional techniques, i.e., situations where it is economical and where the material properties are satisfactory.

As the principal advantage of SLM technology lies in producing small and complex products when the material properties cannot be tested by standardized test specimens, it would be convenient to have another possibility of studying mechanical properties either in the as-received state or even under operating conditions.

One way how to test small size components is to use the small punch test (SPT) method [6]. SPT is a progressive testing method of miniature test samples and is very often used, for instance, in nuclear and fossil fuel power plants for determining residual lifetime of the key pressurized parts. It is applied to determine the mechanism of failure and damage of the equipment too [7–9]. There are many literature references and extensive and long-term experience with this method exploited in practice for wrought steels, but to the author's knowledge, there is only limited information about the use of this method for additive manufactured (3D-printed) materials [10–13].

As is generally known, steel produced by the SLM process contains either more or fewer pores, which have a significant effect on the strength and fracture properties of steel. The steel produced by SLM technology has also a special texture and its microstructure is very similar to welds [5] and the material properties are therefore different in various printing directions [14], including creep [15] and fatigue [16] properties. The papers [15,17,18] show a lower creep service life of materials produced by 3D printing (SLM method) compared to conventional manufacturing.

There is a lot of information in the literature about the influence of porosity, printing (building up) direction, velocity and laser power on the material properties of intentionally prepared test specimens but few about properties of the real products. Works which examine real products are, for instance, concerned with machining issues [19], cutting forces and anisotropy features in the milling [20] and methods for the inspection scheduling of additive manufactured structures [21]. It is known and proven that the microstructure and properties are affected by the cooling rate associated with, among other factors, heat dissipation, which depends on the size of the product as well. This is the reason why we decided to investigate the long-term behavior of austenitic steel on blocks produced by SLM and to study the difference between intentionally produced samples presented in the literature and a massive block produced within this study. This paper thus summarizes and compares the material properties of a block made of AISI 316L steel produced by SLM with a hot-rolled steel plate of the same steel grade. Both materials were tested by conventional testing methods and small punch tests and stress rupture tests were supplemented by the analysis of microstructure.

## 2. Materials and Methods

The experimental material (3D-printed steel) was a block made of austenitic stainless steel AISI 316L with the following dimensions for printing: base, 15 mm × 280 mm and height, 170 mm. The block of 3D-printed steel was cut into two halves, the first stayed in as-printed condition (marked hereafter as 3D) and the second was solution-annealed (marked hereafter as 3DS).

Solution annealing was performed in a laboratory electric furnace at 1050 °C with a holding time of 15 min followed by cooling in water.

The comparative test material was a hot-rolled plate of AISI 316L steel with similar dimensions as the 3D-printed block and the solution annealed when exposed to 1050 °C and cooled in water.

### 2.1. Blank Printing

The block was produced by an additive manufacturing process in the Protolab 3D printing center of VSB—Technical University of Ostrava in Czech Republic on a Renishaw AM400 (Wotton-under-Edge, Great Britain) in the 2019.

The parameters of the selective laser melting (SLM) were chosen by the manufacturer, namely: laser power 200 MW, layer thickness of 50 µm, scanning speed ($v_{scan}$) 650 mm/s and chessboard strategy, Figure 1, and without supporting elements.

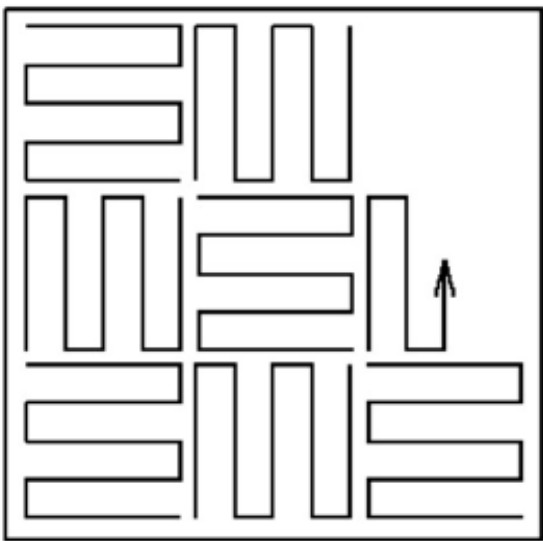

**Figure 1.** Chessboard strategy.

The powder SS 316L-0407 supplied by Renishaw is used for manufacturing, the chemical composition of the powder is provided in Table 1 and the powder properties are examined in detail in the following section [22].

**Table 1.** Chemical composition of Powder SS 316L-0407, 3D-printed block and plate (wt.%).

| Type of Product | C | Mn | Si | P | S | Cu | Ni | Cr | Mo | V | O | N |
|---|---|---|---|---|---|---|---|---|---|---|---|---|
| Powder | ≤0.03 | ≤2.00 | ≤1.00 | ≤0.045 | ≤0.030 | - | 10.00 to 14.00 | 16.00 to 18.00 | 2.00 to 3.00 | - | ≤0.10 | ≤0.10 |
| 3D | 0.025 | 0.490 | 0.660 | 0.0160 | 0.0050 | 0.110 | 12.700 | 16.600 | 2.360 | 0.039 | 0.001 | <0.001 |
| Plate | 0.019 | 1.851 | 0.265 | 0.0372 | 0.0014 | 0.305 | 10.001 | 16.986 | 2.022 | - | - | 0.0298 |

*2.2. Scope of Testing*

Mechanical properties were studied in both conditions, whereas creep behavior was studied only in the as-printed condition. Testing samples were prepared in parallel (L) and perpendicular (T) directions to the printing base (PB), see Figure 2, except for stress rupture tests that were machined only in one direction. The hot rolled plate of the thickness of 16 mm made of the same steel grade (marked hereafter as P) was used for the comparison of the material properties with the 3D-printed block. Samples for mechanical testing were made from the plate in two directions, longitudinally (L) and transversely (T) to the rolling direction, as seen in Figure 2.

Conventional creep test specimens with circular cross-sections were used for the stress rupture testing of SLM steel as well as hot-rolled plate. One test specimen for one testing condition (temperature and stress) was prepared.

*2.3. Test Equipment and Principle of Applicated Methods*

2.3.1. Chemical Composition

Chemical analysis was in both cases performed on X-ray fluorescence spectrometer ARL ADVANTX; the concentration of C, S and N was tested on a LECO CS 230 and LECO TCH 600; and the oxygen content was measured on a PERKIN ELMER OES ICP, OPTIMA 8000.

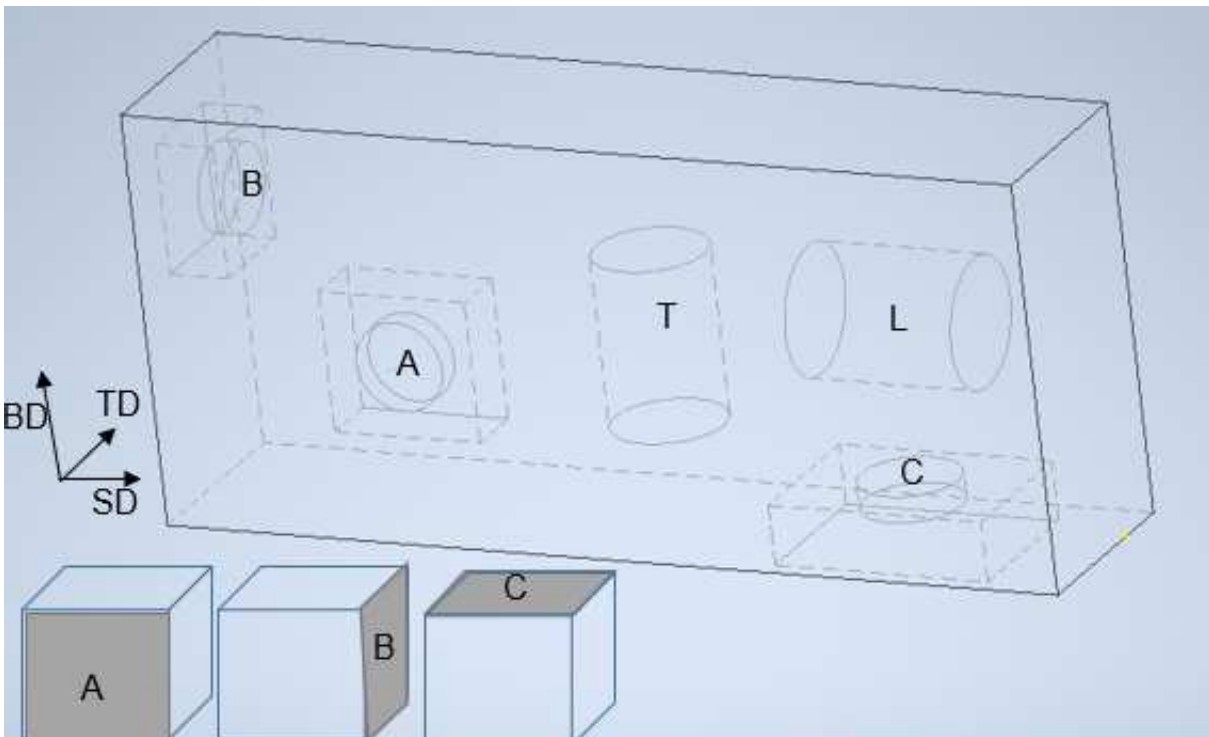

**Figure 2.** Direction of testing, where (L) and (T) stand for the conventional tensile tests; (A)–(C) for SPT tests and microstructure analysis; and BD—building direction, SD—scanning direction and TD—thickness direction.

### 2.3.2. Tensile and Charpy V-Notch Tests

Tensile tests were performed on MTS 500 kN test rig (speed 0.6 mm/min) at laboratory temperatures and the tensile testing was carried out and evaluated in accordance with the standard EN ISO 6892-1.

Charpy tests were performed in accordance with the EN ISO 148-1 standard on specimens 10 × 10 × 55 mm with an ISO-V notch at room temperature. Three samples of the same condition or direction were prepared.

### 2.3.3. Small Punch Tests

Moreover, the conventional tensile tests small punch test (SPT) method is used for the determination of the mechanical properties at room temperature of hot-rolled steel as well as 3D-printed block. The small punch test is based on the penetration of a disc-shaped specimen until it breaks. The method was described in detail elsewhere [23].

Disc-shaped specimens with 8 mm in diameter and 0.5 mm in thickness ($h_0$) were prepared from each direction of the 3D-printed material surface (A), transverse (B) and longitudinal (C), in both as-printed and solution-annealed conditions (Figure 2). The SPT samples of hot-rolled plate were machined only on the surface (A), according to the standard [6]. Five SPT samples for the same condition or direction were prepared and tested in order to obtain the information about the scatter of results.

The principle of the small punch test is the penetration of a punch through the small disc specimen that is clamped between the upper and lower die of the specimen-holder in the testing device. Although the testing standard [6] specifies a punch diameter of 2.5 mm, another punch diameter (2 mm) was used in this experiment because of our database of SPT results and more than ten years of experience based on a 2 mm punch diameter. During the test, the values of force and displacement of the punch tip are recorded and a curve with a typical shape for most of the structural materials tested at room temperature is plotted. The

following characteristics used for estimating the strength and fracture behavior of materials can be determined from this curve:

- $F_e$—the elastic–plastic transition force in the small punch test which characterizes the transition from linearity to the stage connected with the development of plastic deformation through the whole thickness of the sample. This value corresponds to the yield strength in the conventional tensile test and is defined as the point of intersection of two constructed tangents (one of initial stiffens and a second of steady-state plastic stretching) (N);
- $F_m$—the maximum force during the test which corresponds to the load at the tensile strength in the conventional tensile test (N);
- $d_m$—the displacement of the punch tip which corresponds to the force $F_m$ (mm).

The guidelines for the translation of SPT data into tensile properties have been established, for instance in [6].

### 2.3.4. Stress Rupture Test Method

Stress rupture testing is the method of uncovering the creep behavior of materials. Stress rupture properties were tested on round test specimens, according to EN ISO 204, on lever-type creep machines with a vertical load axis. Specimen elongation is not continuously recorded in this type of test and the load is kept constant until the specimen ruptures. Stress rupture tests were performed at temperatures 650 and 700 °C in the stress range that should cover rupture times of approximately 100, 200, 500, 1000, 2000 and 5000 h.

The testing parameters were almost the same for both 3D-printed blocks and the plate, the last two tests of the plate at 650 °C are still running and are not stated here.

### 2.3.5. Fractographic and Metallographic Investigation

Fractographic investigation was performed on the fracture surface of some of the 3D and 3DS Charpy V-notch test, SPT test and stress rupture test specimens by using the scanning electron microscope JEOL JSM-5510 equipped with the EDX microanalyzer X-max 20 (Oxford Instruments, Oxford, UK).

The microstructure of 3D and 3DS samples was analyzed in three directions, on the surface of the block (A), perpendicular to the printing base (B) and parallel to the printing base (C), as seen in Figure 2. The metallographic samples were prepared by grinding, polishing and etching in a V2A etchant and analyzed on a OLYMPUS GX51 inverted metallographic microscope (Shinjuku, Japan).

## 3. Results

### 3.1. Chemical Composition

The chemical composition of the steel powder, the 3D-printed block and hot-rolled plate is stated in Table 1. The chemical composition of the 3D-printed block completely conformed the requirements for the steel AISI 316L with a low concentration of trace elements and even gases (O, N). The hot-rolled plate had slightly lower carbon, manganese, silicon and nickel contents than 3D-printed block but all is still in accordance with the material standards.

### 3.2. Tensile Test and Charpy V-Notch Test

The results of tensile test and Charpy V-notch test performed at room temperature are stated in Table 2. The principal difference between 3D-printed and hot-rolled material is in yield stress where it is more than 100 MPa higher in 3D-printed block after solution annealing and even twice as high in 3D as-printed block compared to hot-rolled plate. The difference in the tensile strength is, on the other hand, quite insignificant. While the material properties of hot-rolled plate are practically identical regardless to the direction, there is a drop in both the yield stress and tensile strength in transverse compared to the longitudinal direction in 3D-printed block in both heat treatment states. The pronounced decrease was detected also in plasticity and impact toughness, when the elongation of

3D-printed material was only about half the value in longitudinal direction and even only a quarter in the transverse direction compared to the hot-rolled plate. The annealing solution then further lowered mechanical properties (yield stress as well as tensile strength), and even reduced the reduction of area.

**Table 2.** Results of mechanical tests.

| | Material | | 3D | | 3D S | | P | |
|---|---|---|---|---|---|---|---|---|
| | Direction | | L | T | L | T | L | T |
| **Mechanical properties** | $R_{p0,2}$ | (MPa) | 557 | 473 | 373 | 352 | 259 | 258 |
| | $R_m$ | (MPa) | 690 | 591 | 621 | 532 | 644 | 643 |
| | A | (%) | 46.5 | 22.0 | 46.5 | 22.5 | 79.0 | 76.0 |
| | R.A. | (%) | 59.5 | 35.0 | 46.5 | 30.5 | 80.5 | 81.5 |
| | KV | (J) | 82 | 67 | 91 | - | 394 | 298 |

The results also show that the 3D-printed material reached only 30% of the absorbed energy of the hot-rolled plate in the Charpy V-notch test and there was hardly any increase in notch toughness after the annealing of the 3D-printed block.

*3.3. Small Punch Tests*

The results of SPT confirmed that the 3D-printed material had higher yield stress and displayed comparable tensile strength to the hot-rolled plate of the same steel, as seen in Figure 3a.

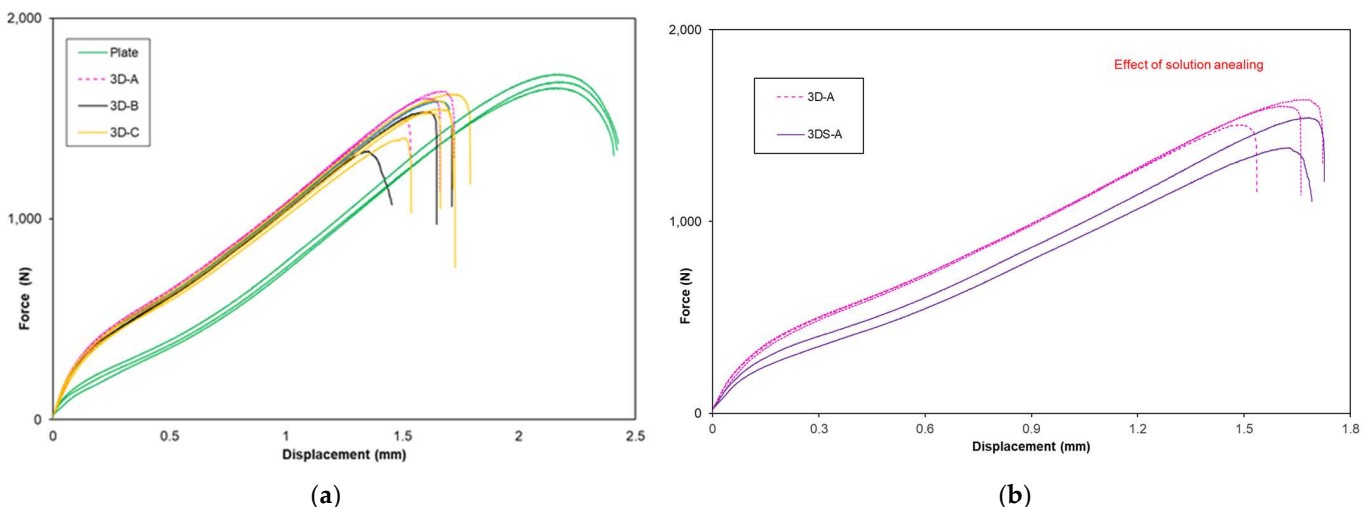

(**a**)　　　　　　　　　　　　　　　　(**b**)

**Figure 3.** SPT load–displacement curves: (**a**) differences between 3D and plate; (**b**) effect of solution treatment on 3D-printed material.

The area under the force–displacement curve represents the fracture energy and can be used for the qualitative estimation of fracture resistance. This area is lesser for 3D-printed steel than for the hot-rolled plate, as seen in Figure 3a, and even lower for the solution annealing conditions of the 3D-printed block (Figure 3b). The decrease of the fracture energy of the 3D material versus plate is about 22%. SPT curves also show a large scatter of results, especially for samples oriented perpendicularly to the printing base (B). On the other hand, the smallest scattering of results was found in samples oriented parallel to the plate surface (A), with the same orientation as the samples scooped in situ from the real surface of an in-service component (short transverse direction), which gives good prospects towards the future in-service testing of real components.

Strength characteristics using SPT are determined on the basis of correlation between the results of conventional tensile tests and small punch tests, according to Equations (1) and (2):

$$R_{p0,2} = \alpha \cdot \frac{F_e}{h_0{}^2} = 0.5564 \cdot \frac{F_e}{h_0{}^2} \qquad \text{(Mpa)} \qquad (1)$$

$$R_m = \beta \cdot \frac{F_m}{h_0 \cdot d_m} = 0.4089 \cdot \frac{F_m}{h_0 \cdot d_m} \qquad \text{(Mpa)} \qquad (2)$$

where $R_{p0,2}$ and $R_m$ are the yield stress and tensile strength, respectively, $h_0$ is the exactly measured thickness of each disc-shaped specimen before testing in mm, $F_e$ is the elastic-plastic transition force in the small punch test in N, $\alpha$ is correlation coefficient, $F_m$ is the maximum force during the SPT in N and $d_m$ represents the displacement of the punch tip in mm, which corresponds to the force $F_m$.

Figure 4 describes these correlations between tensile tests and SPT valid for austenitic stainless tubes tested at MMV and is supplemented by the results of the tested hot-rolled plate and 3D-printed block. It is clear that the last results lie below the respective correlations.

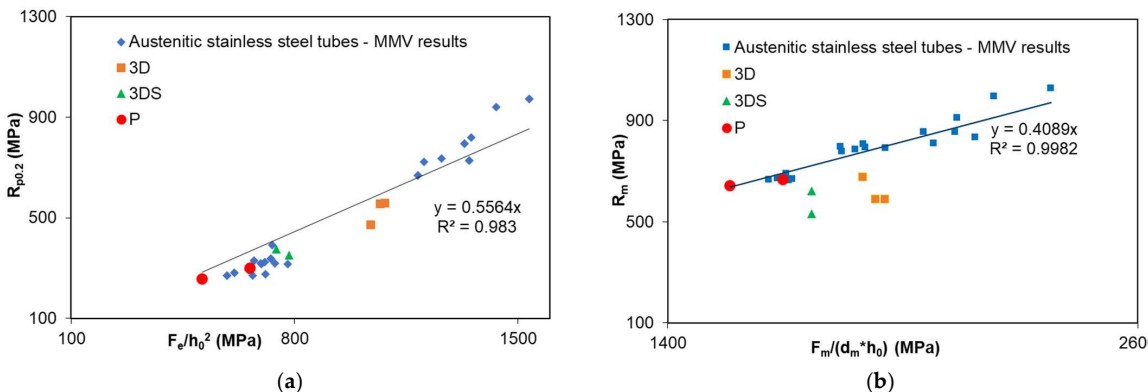

(a)                        (b)

**Figure 4.** Correlation of SPT and standard tensile test data for austenitic steels the results present: (**a**) yield point, (**b**) tensile strength.

Figure 5 compares the SEM images of the ruptured SPT specimens. While in the hot-rolled sample the crack was circumferential, as is usual in metallic materials (Figure 5a), there were several branched cracks and othar point defects on the surface of the 3D sample, as seen in Figure 5b.

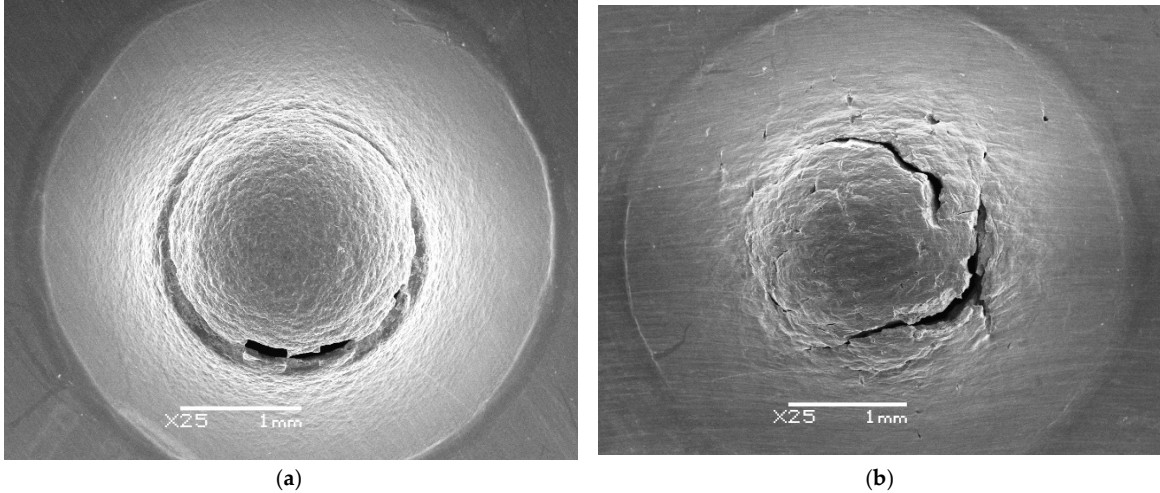

(a)                        (b)

**Figure 5.** SEM images of macroscopic cracking morphology of the small punch test samples of the hot-rolled plate (**a**) and 3D-printed block (**b**).

### 3.4. Stress-Rupture Tests

The comparison of the creep resistance of both the 3D-printed block (samples No. 1–10, Table 3) and hot-rolled plate (samples No. 11–20, Table 4) was performed by stress rupture tests at 650 and 700 °C.

**Table 3.** Results of stress rupture tests of selective laser melting AISI 316L steel plate.

| Sample No. | Temperature °C | Stress Mpa | Time to Rupture h | Elongation % | R.A. % |
|---|---|---|---|---|---|
| 1 | 650 | 130 | 4576 | 16.4 | 11.3 |
| 2 | 650 | 150 | 865 | 14.2 | 6.8 |
| 3 | 650 | 170 | 216 | 10.0 | 7.9 |
| 4 | 650 | 190 | 311 | 13.9 | 7.1 |
| 5 | 650 | 220 | 52 | 14.8 | 8.3 |
| 6 | 700 | 80 | 2384 | 20.2 | 10.8 |
| 7 | 700 | 95 | 1435 | 13.4 | 8.9 |
| 8 | 700 | 115 | 884 | 18.1 | 8.1 |
| 9 | 700 | 130 | 338 | 13.6 | 14.2 |
| 10 | 700 | 155 | 103 | 16.0 | 13.6 |
| 11 | 700 | 175 | 46 | 14.0 | 14.5 |

**Table 4.** Results of stress rupture tests of the hot-rolled AISI 316L steel plate.

| Sample No. | Temperature °C | Stress Mpa | Time to Rupture h | Elongation % | R.A. % |
|---|---|---|---|---|---|
| 12 | 650 | 170 | 1863 | 19.2 | 68.6 |
| 13 | 650 | 190 | 523 | 16.8 | 64.8 |
| 14 | 650 | 220 | 35 | 50.3 | 57.8 |
| 15 | 650 | 240 | 11 | 43.9 | 56.4 |
| 16 | 700 | 65 | 4967 | 39.6 | 69.3 |
| 17 | 700 | 80 | 3288 | 44.8 | 66.1 |
| 18 | 700 | 95 | 1245 | 28.8 | 65.0 |
| 19 | 700 | 115 | 1234 | 38.8 | 70.2 |
| 20 | 700 | 130 | 598 | 32.2 | 78.3 |
| 21 | 700 | 155 | 205 | 19.6 | 68.3 |
| 22 | 700 | 175 | 12 | 39.2 | 70.2 |

The results are plotted in Figure 6 in terms of the stress vs. the rupture time. It was confirmed that the creep rupture life, $t_f$, and the stress, $\sigma$, can be described by the Norton equation:

$$t_r = A_r \cdot \sigma^{-n_r} \tag{3}$$

where $A_r$ and $n_r$ are the temperature-dependent constants summarized in Table 5 and in Figure 6a,b.

Regardless of the low stress dependence of time to rupture at 650 °C of the hot-rolled plate, which can be due to only restricted results and their relatively short times to rupture, the results of stress rupture tests of both tested materials are very well comparable.

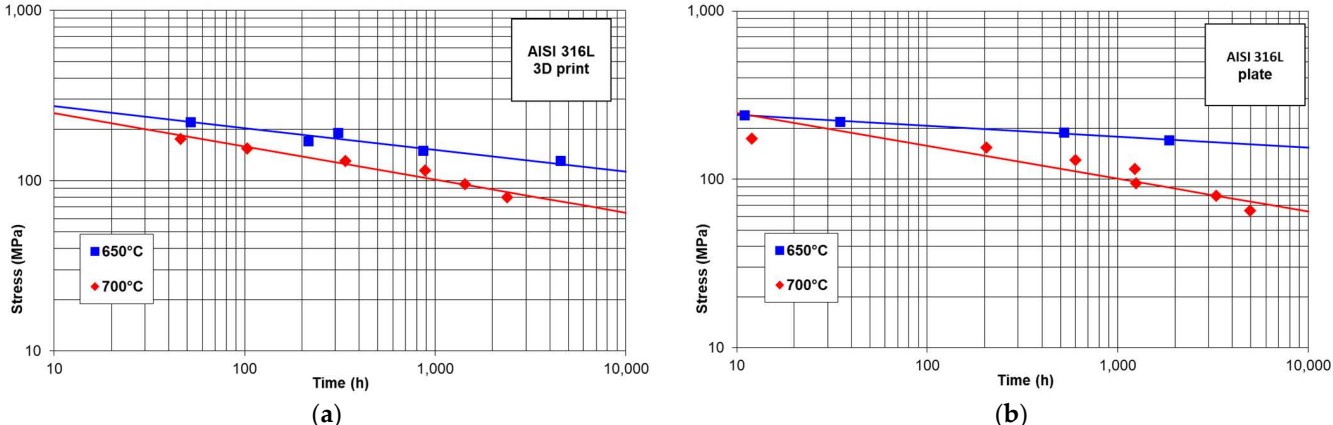

**Figure 6.** Comparison of the results of stress rupture tests of the 3D block (**a**) and hot-rolled plate (**b**). Lines represent the corresponding Norton equations.

**Table 5.** Constants of Norton equation for the 3D and hot-rolled material of AISI 316L steel.

| Material | Temperature (°C) | $A_r$ | $n_r$ |
|---|---|---|---|
| 3D block | 650 | $1.20 \cdot 10^{20}$ | 7.83 |
| | 700 | $2.02 \cdot 10^{13}$ | 5.14 |
| Hot-rolled plate | 650 | $5.00 \cdot 10^{37}$ | 15.41 |
| | 700 | $1.92 \cdot 10^{13}$ | 5.13 |

In order to compare both tested materials and to confront the results of stress rupture tests with the data of the material standard of AISI 316 steel, results were also recalculated by using the Larson–Miller parameter, which allows the combination of the temperature and time to rupture in the form of:

$$P_{LM} = T \cdot [C + \log(t)] \tag{4}$$

where *T* is temperature in Kelvin and *t* is the time to rupture in hours. The value of the Larson–Miller parameter (*C* = 15.2) was calculated using the least square method from the mean values of the creep rupture strength, as stated in the material standard EN 10 216-5 for steel X6CrNiMo17-13-2 (AISI 316H).

The comparison shows that the results of stress rupture tests, however short-lived and without regarding the above-shown differences in stress–time to rupture dependence, lie within the range between the mean value of CRS (solid line in Figure 7) and −20% tolerance limit (dashed line in Figure 7) of the steel 316. Although the obtained results of the stress rupture test do not allow the estimation of CRS in 10,000 h so far, it can be seen that both the hot-rolled plate and 3D-printed block have fairly good prospects for achieving good creep properties.

*3.5. Analysis of Microstructure and Fracture Surfaces*

The highly localized melting, strong temperature gradient and a high solidification front rate, which are associated with the 3D printing of metallic materials, generate a microstructure with extreme nonequilibrium, far from that accessible through conventional manufacturing methods. The microstructure analysis of the 3D-printed samples in the as-printed state (3D) and after solution annealing (3DS) was performed in three directions, on the surfaces A, B and C (Figure 2).

The microstructure of the studied samples developed by etching in V2A revealed the microstructure typical for 3D-printed metals, i.e., multilayered structure with clearly detectable strip boundaries (molten pools). Grains with a dendritic arrangement and

precipitates along grain boundaries were observed (Figures 8 and 9). Solution annealing led to the dissolution of the strip boundaries, but the influence of build orientation remained preserved even after solution annealing (Figure 10).

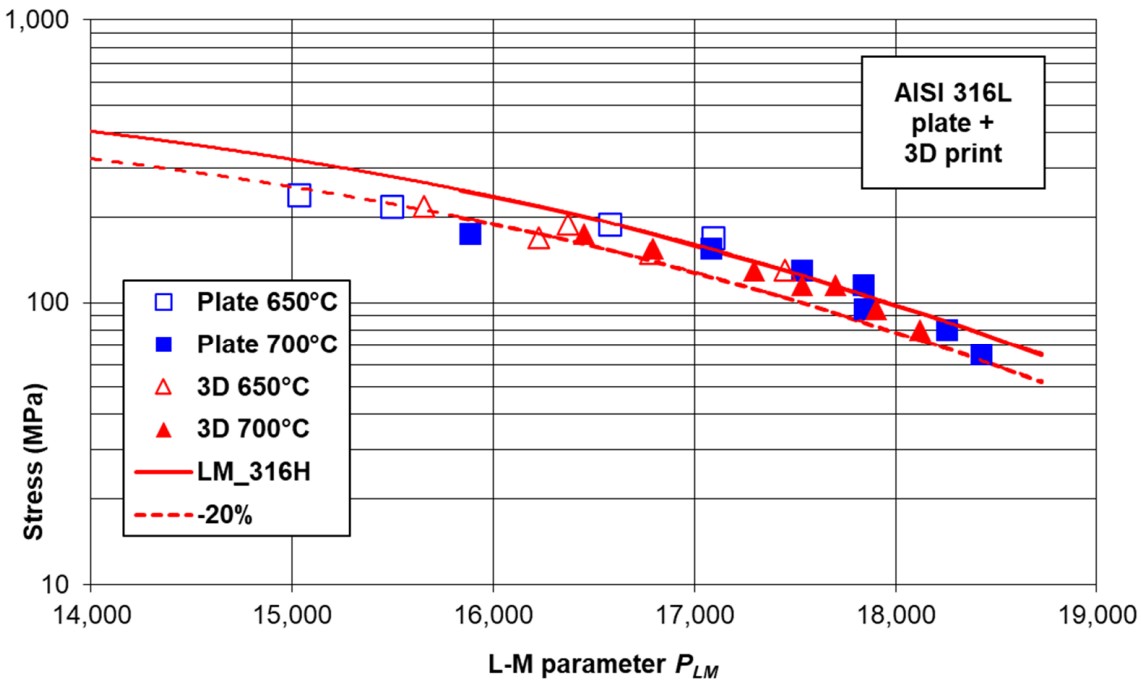

**Figure 7.** Comparison of the results of stress rupture tests of the 3D-printed block and hot-rolled AISI 316L steel plate with the mean value and −20% tolerance limit of creep rupture stress (CRS) for AISI 316H steel.

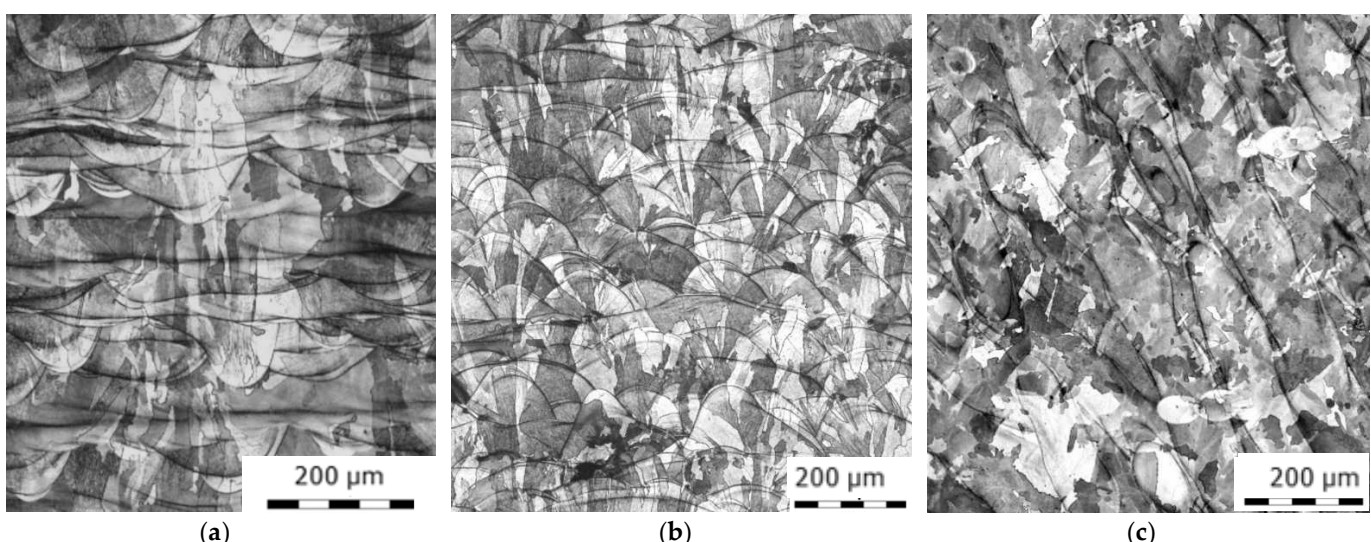

**Figure 8.** Microstructure of 3D samples: (**a**) surface A; (**b**) perpendicular B; (**c**) parallel C to the printing base.

The variety of size and shapes of the austenitic grains in the 3D-printed block was far from the equiaxed austenitic grains, with annealing twins found on the hot-rolled plate, as seen in Figure 11.

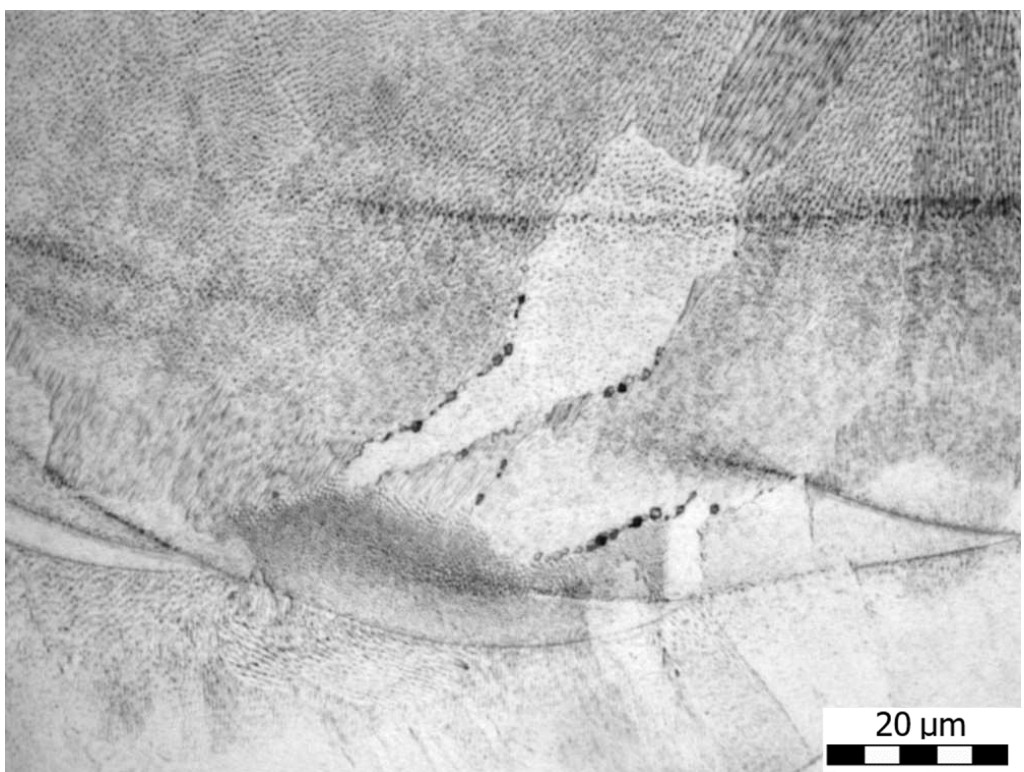

**Figure 9.** Dendritic arrangement of microstructure and precipitates along grain boundaries (sample 3D).

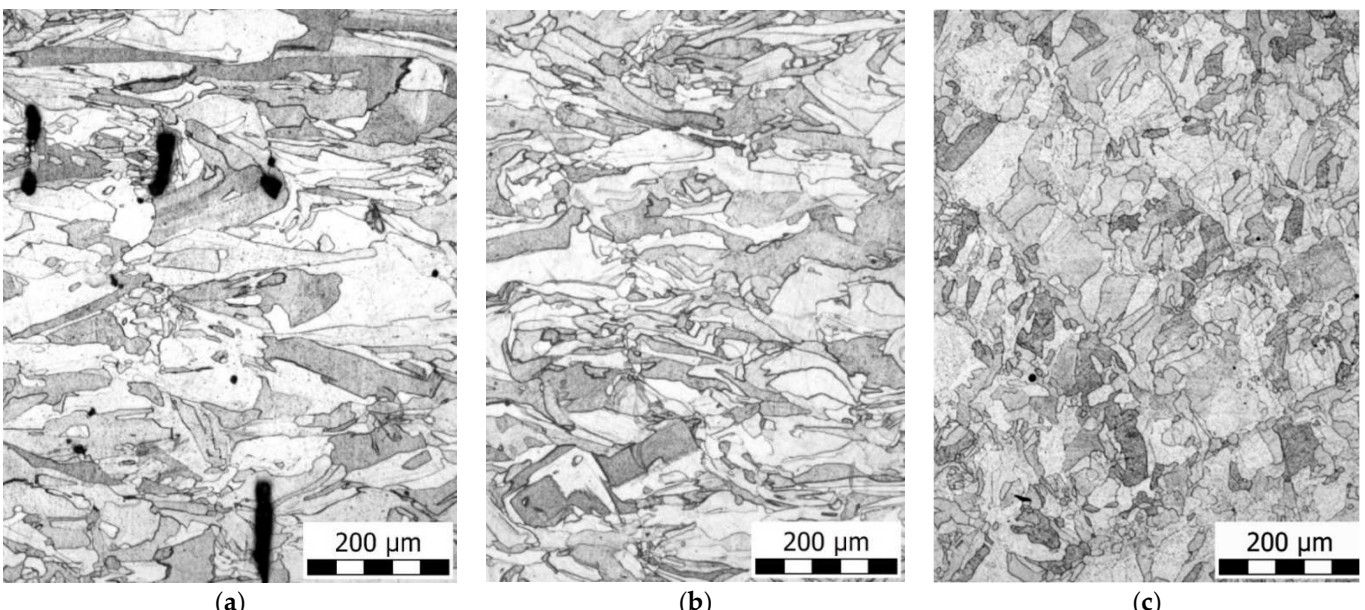

| (a) | (b) | (c) |

**Figure 10.** Microstructure of 3DS samples: (**a**) surface A; (**b**) perpendicular B; (**c**) parallel C to the printing base.

The mechanical properties and, especially, the toughness of the steel always reflect the state of the microstructure and the presence of defects. The comparative fractographic analysis of Charpy V-notch test samples of the 3D-printed block and hot-rolled plate confirmed fully the transgranular ductile fracture with typical elongated dimples which were very fine in the case of 3D-printed material with the size corresponding with the fine cellular structure and were relatively coarse in the hot-rolled plate (compare Figures 12a, 13a and 14). However, the fracture surface of 3DS samples was a combination

of two mechanisms where the transgranular-dimpled ductile fracture prevailed but partly appeared as a transgranular cleavage fracture (Figure 13a). A large number of defects, such as dilutions, and unmelted metal particles, such as balls (blue arrows in Figure 12b), were detected on the fracture surfaces of both 3D and 3DS-printed material variants, see Figures 12b and 13b.

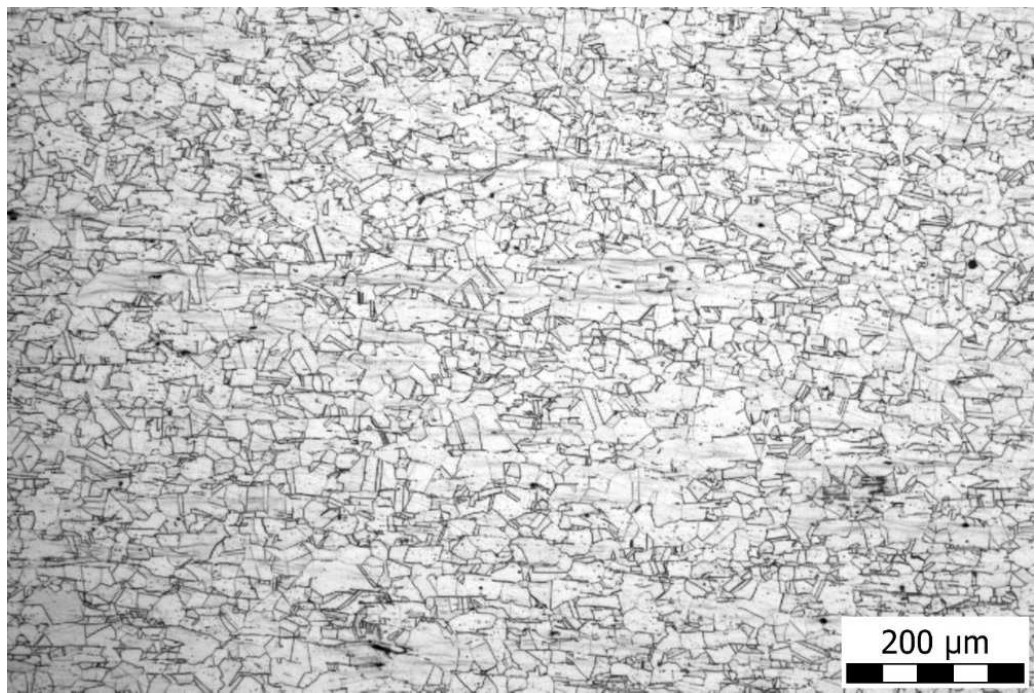

**Figure 11.** Microstructure of hot-rolled austenite plate.

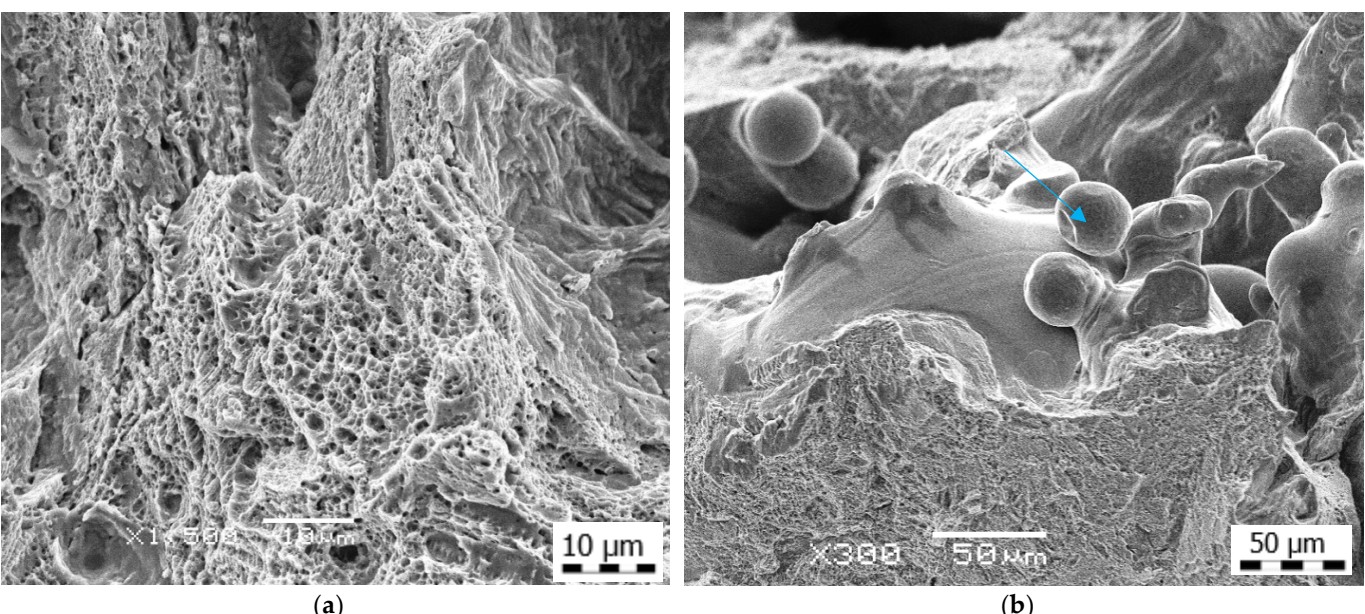

(**a**)                                          (**b**)

**Figure 12.** Fracture surface of 3D Charpy V-notch test: (**a**) transgranular fracture; (**b**) defects on the fracture surface, blue arrow shows unmelted metal particles.

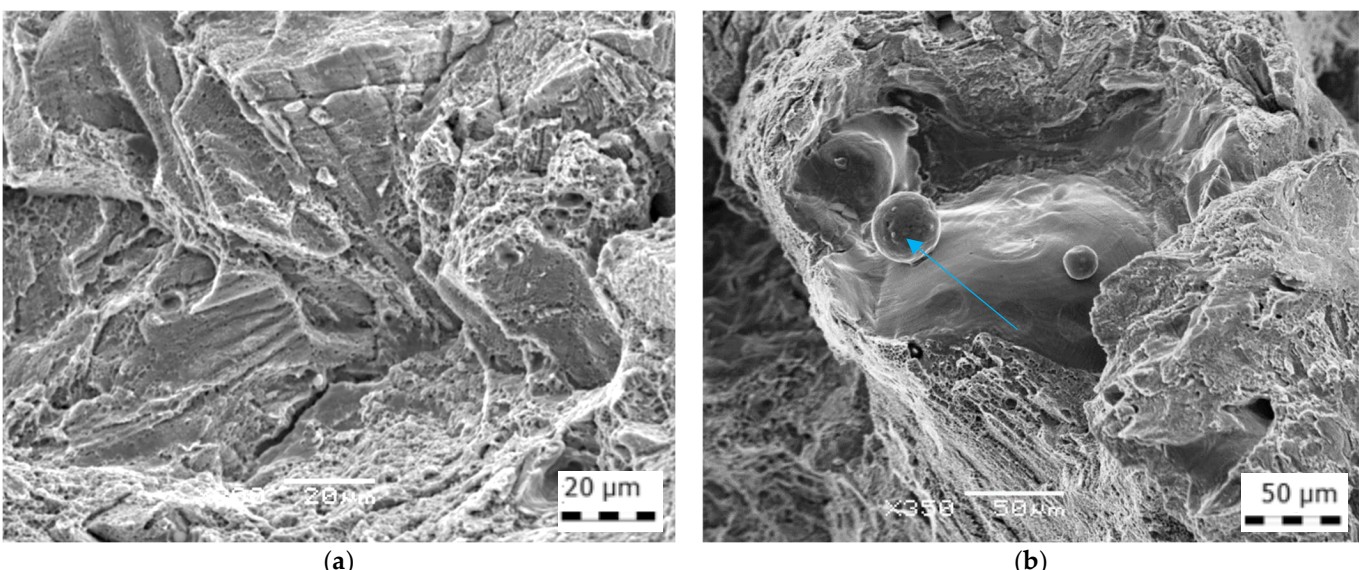

**Figure 13.** Fracture surface of 3DS Charpy V-notch test: (**a**) combination of the fracture mechanisms; (**b**) defects on the fracture surface, blue arrow shows unmelted metal particle.

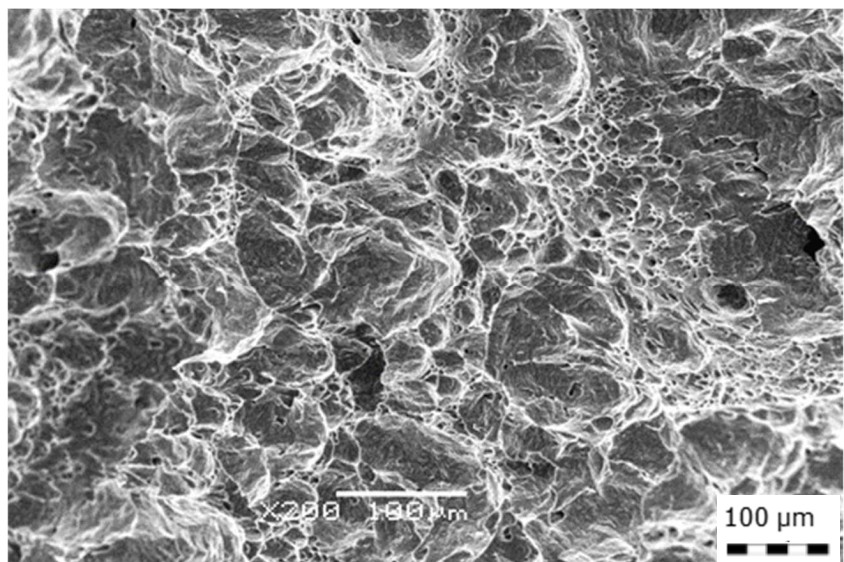

**Figure 14.** Fracture surface of hot-rolled plate.

## 4. Discussion

The results of mechanical tests, including SPT, show that the 3D-printed material has higher yield stress and a comparable tensile strength to the hot-rolled plate of the same steel but significantly lower plasticity and fracture toughness. Such behavior is a result of the printing technology, which is characterized by the special structure that looks like fine molten pools arranged in bands that are contoured with the columnar grains, inside which a fine cellular structure appears. Low plasticity is then due to numerous defects and imperfections, such as pores and unmelted powder. This special combination of material properties and microstructure is a typical feature of 3D-printed materials [14]. The microstructure and also the strength and plasticity can be improved by hot isostatic pressing, as is recommended in [24]. Our results did not confirm the pronouncedly positive effect of heat treatment, especially on ductility.

The defects in microstructure overcame the positive effect of the fine grain size and the presence of the substructure in the 3D-printed block where the size of dimples in the

fracture surface was several times smaller than in the hot-rolled plate and reduced the plasticity in both tensile as well as stress rupture tests.

When we compare mechanical properties (yield strength and tensile strength) of our 3D-printed steel, with the results reported in [25], the strength is higher, about 80 MPa in the scanning direction (15%) and about 45 MPa in the building direction (10%). The effect of the test direction was maintained, as well as the yield strength/tensile strength ratio (0.80), which is twice as high for 3D-printed steels than for wrought austenitic stainless steels (about 0.40).

We assume that our results were influenced by the production of a large block (dimensions $170 \times 250$ mm), which generated higher internal stress and the accumulation of inhomogeneities in the material compared to the production of intentionally manufactured samples or test specimens, which have been widely studied. This assumption is also confirmed by the results obtained on half of the 3D-printed block that was solution-annealed, which led to the elimination of melting bands, the partial homogenization of properties, the reduction in yield strength (YS) without changes in ductility (Figure 15) and the slight increase in notch toughness. Although this heat treatment most likely reduced the stress, as evidenced by the decrease in yield strength, it did not remove unmelted powder and pores, so there was no detectable improvement in plasticity after solution annealing. The results of the stress rupture tests are comparable with the data published by other authors [26–29], see Figure 16. At the same time, all data lie close to the mean CRS stated in the material standard of AISI 316H, which is the closest heat-resistant equivalent to the steel AISI 316L. All experimental data stated in Figure 16 are taken from the vertical samples (the long axis of the sample is parallel with the building direction) and the results show that the creep resistance of the 3D-printed material is not negatively influenced by the size of the printed block. Figure 15 compares our results with literature values from [14,30–34].

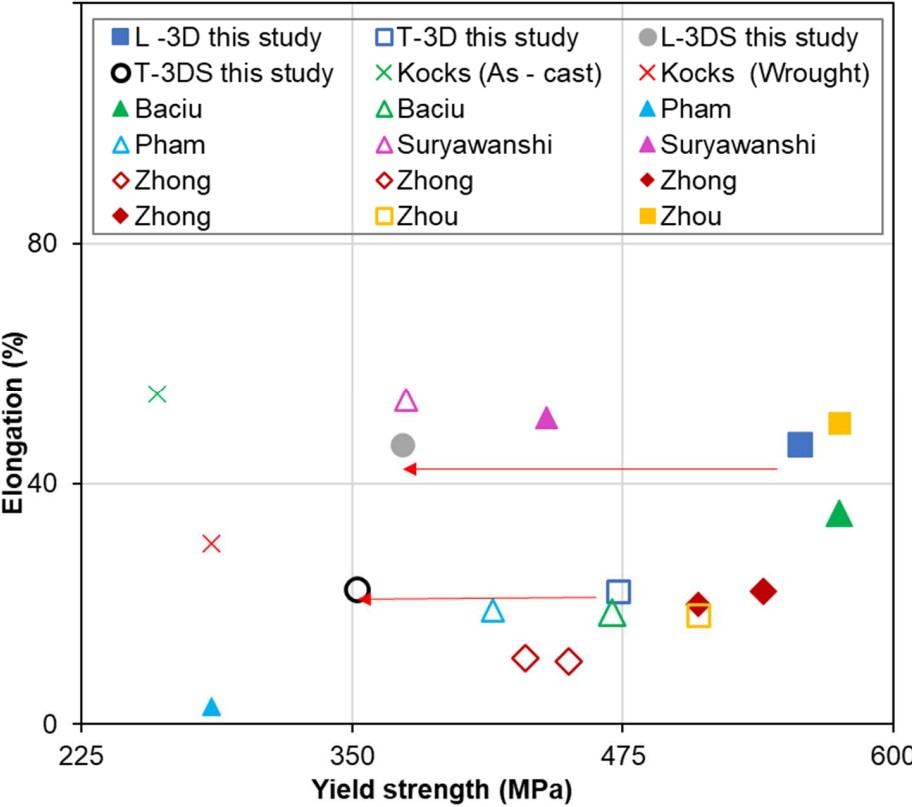

**Figure 15.** Relation between elongation and yield strength from literature and this study, red arrows show the effect of solution annealing.

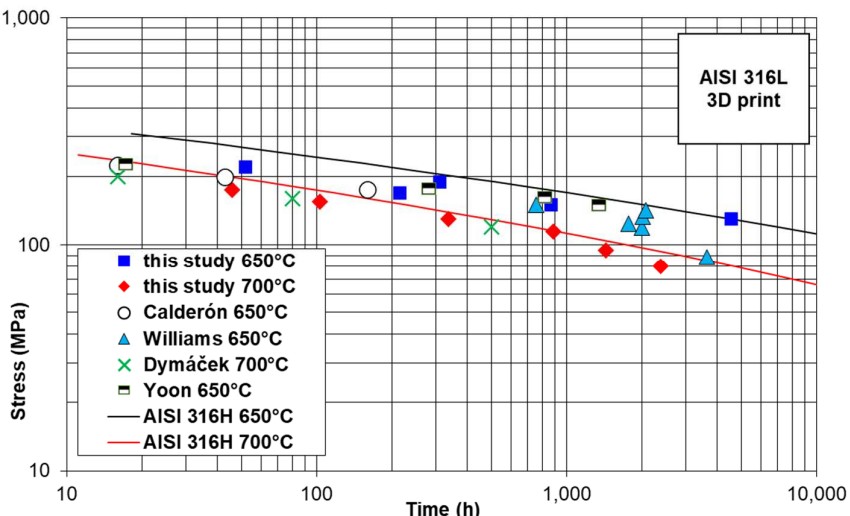

**Figure 16.** Comparison of stress–time to rupture dependence for 3D-printed steel 316L, lines represent the mean CRS of AISI 316H steel.

As both dislocation and diffusion creep rely on the diffusion of atoms and the grain boundaries represent the fast diffusion path, it is clear that the creep rate and reciprocal time to rupture strongly depends on the grain size: the smaller the grain size, the larger grain boundary areas and the faster diffusion rate. Diffusion is also dominant in the stress direction as the atomic spacing and vacancies are increased. Columnar grains in 3D-printed material grow in the building direction. The vertically-built specimens are therefore loaded parallel to the direction of columnar grain growth and the maximum principal stress is also parallel to the grain boundaries. In this case, the grain size is relatively large along the direction of loading compared to the horizontal case, and diffusion paths are longer. Vertical specimens should thus exhibit longer times to rupture compared to horizontal ones [27] and could have the closest creep resistance to the wrought material. The results of Calderón [26] and this work seem to confirm this, at least from time to rupture up to 10,000 h (Figure 17).

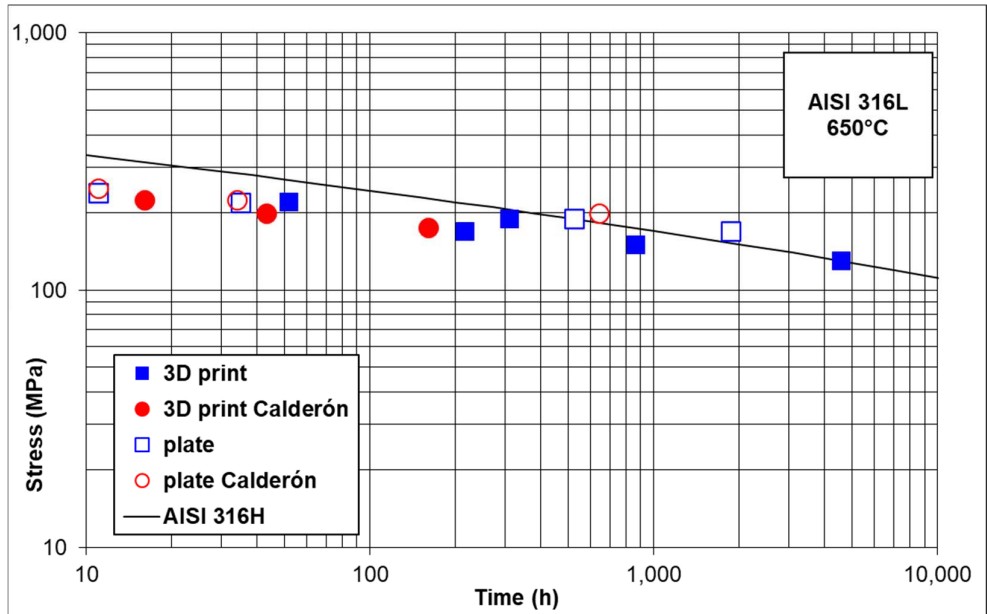

**Figure 17.** Comparison of stress–time to rupture dependence for 3D-printed and hot-rolled steel AISI 316L, lines represent mean CRS of AISI 316H steel at 650 °C.

The difference between 3D-printed and hot-rolled material fits well within the usual scatter band of the results of creep tests and the results are quite even close to the mean value of CRS for the steel AISI 316H.

The obtained results also showed that the correlation curves routinely applied in small punch testing of wrought austenitic steels for determining tensile strength cannot be used for defining the tensile strength (UTS) of the 3D-printed material, because there is high value dispersion and therefore the prediction of the UTS value can be misleading, see Figure 4b. On the other hand, the use of such a correlation for the estimation of YS is appropriate, because the variance is minimal, and the coefficient of reliability is high and almost reaches the value valid for wrought stainless steels (Figure 4a).

### 5. Conclusions

This paper summarized the results of the study focused on the influence of additive manufacturing on the material properties of AISI 316L steel with the same dimensions as the hot-rolled plate made of the same steel.

- The results of tensile-testing at room temperature confirmed the much higher yield stress of 3D-printed material compared to the hot-rolled plate of the comparable tensile strength due to fine cellular structure and its as-welded state. It was, at the same time, accompanied by a drop in elongation as well as impact energy.
- The solution annealing of the 3D-printed block led to the homogenization of properties but could not reduce the pores and particles of unmelted metal powder.
- The good mechanical properties of a 3D-printed block also confirmed the results of the comparative stress rupture test with the hot-rolled plate when the results of both test series were very well comparable and lay in the scatter band around the mean values of the creep rupture stress for the steel AISI 316.

**Author Contributions:** Conceptualization, Š.H.; mechanical testing, P.Č.; metallography analysis, J.K.; fractography, G.R.; small punch testing, O.D.; stress rupture testing Z.K. and G.R.; heat treatment, M.C.; resources, Š.H. and Z.K.; writing—original draft preparation, Š.H.; writing—review and editing, Š.H. and Z.K. All authors have read and agreed to the published version of the manuscript.

**Funding:** This work was supported by the EU Regional Development Fund within the Operational Program Research, Development and Education under the aegis of Ministry of Education, Youth and Sports of the Czech Republic; project number CZ.02.1.01/0.0/0.0/17_049/0008399.

**Institutional Review Board Statement:** Not applicable.

**Informed Consent Statement:** Not applicable.

**Data Availability Statement:** Not applicable.

**Conflicts of Interest:** The authors declare no conflict of interest.

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
