# Peer review of "Study of Material Properties and Creep Behavior of a Large Block of AISI 316L Steel Produced by SLM Technology"

_metals, doi:10.3390/met12081283_

Round 1

Reviewer 1 Report

The Introduction is not well written. The Introduction contains mainly a description of the technology, as in a textbook, but should contain brief research results on your work topic, precisely what was done on the topic of work before your article.

In the Introduction, you write about the almost complete absence of work on the study of the properties of parts and materials obtained using additive technologies. At the same time, there are studies devoted not only to porosity but also to the mechanical properties of materials. These works often indicate that the properties of parts printed by SLM and EBW lie between the properties of cast and forged parts. Therefore, in your case, it is appropriate to say that the results of the study of creep, and not all mechanical properties in general, are poorly presented.

The Methodology section needs some work. It would be good to give the chemical composition of the steel you printed from, the characteristics of the powder (make the reference that the composition is given in Table 1 in the next section), which was printed, printing modes, and all the equipment for printing and mechanical testing. Specify the make, model, manufacturer, and country of this equipment. For better understanding, divide this section into subsections (for example, blank printing, mechanical testing, and structural studies).

In Figure 12 (a and b), you have rulers in different styles, in the left figure, the ruler blends into the white background of the caption. Make rulers and captions in the same style and contrast with the background.

In general, in the section Results and Discussion, you have presented a lot of results of experimental work. Dependences of properties on a manufacturing method are shown. There is a discussion of the reasons for the formation of such properties. I would like to slightly expand the conclusions and add more information about creep as a novelty of this work. It would be useful, in the Conclusion, to indicate in percent the difference in the properties of printed products from the properties of samples obtained by other methods.

Author Response

Dear Reviewer 1

 Comments:

The Introduction is not well written. The Introduction contains mainly a description of the technology, as in a textbook, but should contain brief research results on your work topic, precisely what was done on the topic of work before your article.

I changed it, but about the large block are no other works.

In the Introduction, you write about the almost complete absence of work on the study of the properties of parts and materials obtained using additive technologies. At the same time, there are studies devoted not only to porosity but also to the mechanical properties of materials. These works often indicate that the properties of parts printed by SLM and EBW lie between the properties of cast and forged parts. Therefore, in your case, it is appropriate to say that the results of the study of creep, and not all mechanical properties in general, are poorly presented.

I corrected it.

The Methodology section needs some work. It would be good to give the chemical composition of the steel you printed from, the characteristics of the powder (make the reference that the composition is given in Table 1 in the next section), which was printed, printing modes, and all the equipment for printing and mechanical testing. Specify the make, model, manufacturer, and country of this equipment. For better understanding, divide this section into subsections (for example, blank printing, mechanical testing, and structural studies).

I corrected it according to your opinion. I completed the data about powder and 3D printer. I divided Section methodology into subsections.

In Figure 12 (a and b), you have rulers in different styles, in the left figure, the ruler blends into the white background of the caption. Make rulers and captions in the same style and contrast with the background.

I corrected them.

In general, in the section Results and Discussion, you have presented a lot of results of experimental work. Dependences of properties on a manufacturing method are shown. There is a discussion of the reasons for the formation of such properties. I would like to slightly expand the conclusions and add more information about creep as a novelty of this work. It would be useful, in the Conclusion, to indicate in percent the difference in the properties of printed products from the properties of samples obtained by other methods.

It is corrected, especially creep is expanded.

Sincerely

Šárka Hermanová et. all

Reviewer 2 Report

This paper provides an laboratory assessment of AISI 316L steel block manufactured by SLM technology. Two different heat treatment was conducted: as-printed and solution annealed. A good literature review including very recent articles published in the open literature domain was conducted to justify the need for this research. The experimental is somewhat clear, however, there may be some misinterpretation due to poor English.  It was concluded that a much higher yield strength was obtained for the SLM 3D printed AISI 316L block in compared with the conventional hot rolled block. The conclusion was drawn based on the experimental investigation. The study is interesting and it will attract both readers in this field and citation. However, the paper needs revision before publication.

1. The English needs to be improved. It starts from the very beginning. All the affiliation seems the same but 7 different affiliations were included with the same details. There are many more grammatical issues throughout the paper.

2. Abstract: It needs to be rewritten including the 3Ps (purpose, procedure and principle findings)and 1C (Context) criteria. Also remove any references from abstract and conclusion.

3.  The introduction is good which includes most recent published articles. However, please avoid grouped citation such as [10-13].

4.  Experimental section: What are the mesh size of the powders? Please include the properties of the powders according the the powder metallurgy description.

5. Figure 1: Write down the full form of SD, TD and BD. Also A, B, C, L and T. Also add the dimensions of h0 etc.

6. What was the laying direction of the block? How many samples for each conditions were tested?

7. Table 3: Why only four samples at 650 and 6 samples at 700 degree centigrade? Did you tested only one sample for each condition? Is the process reproducible? These information need to be added before hand in the experimental section.

8. Table 4: the similar problem. why only 3 samples at 650?

9. Figure 8 should be merged with figure 1.

10.  Microstructural analysis was good. However, in figure 13 a very unusual scale of 13 micro meter was used. Please double check this.

7. 

Author Response

Dear Reviewer 2

This paper provides an laboratory assessment of AISI 316L steel block manufactured by SLM technology. Two different heat treatment was conducted: as-printed and solution annealed. A good literature review including very recent articles published in the open literature domain was conducted to justify the need for this research. The experimental is somewhat clear, however, there may be some misinterpretation due to poor English.  It was concluded that a much higher yield strength was obtained for the SLM 3D printed AISI 316L block in compared with the conventional hot rolled block. The conclusion was drawn based on the experimental investigation. The study is interesting and it will attract both readers in this field and citation. However, the paper needs revision before publication.

  1. The English needs to be improved. It starts from the very beginning. All the affiliation seems the same but 7 different affiliations were included with the same details. There are many more grammatical issues throughout the paper.
    We revised english, affiliations were changed.

2. Abstract: It needs to be rewritten including the 3Ps (purpose, procedure and principle findings)and 1C (Context) criteria. Also remove any references from abstract and conclusion.

We change the abstract according to recommendation.

3.  The introduction is good which includes most recent published articles. However, please avoid grouped citation such as [10-13].

New literature was added, but it is difficulty or not possible to avoid grouped citations for some text. 

4.  Experimental section: What are the mesh size of the powders? Please include the properties of the powders according the the powder metallurgy description.

It was added information about powder and literature where is study about the powder.

5. Figure 1: Write down the full form of SD, TD and BD. Also A, B, C, L and T. Also add the dimensions of h0 etc.

Some of them are in article before the Figure and  I added other.

6. What was the laying direction of the block? How many samples for each conditions were tested?

This information is added.

7. Table 3: Why only four samples at 650 and 6 samples at 700 degree centigrade? Did you tested only one sample for each condition? Is the process reproducible? These information need to be added before hand in the experimental section.

There are some long-term stress rupture tests still running (mainly plate) but they have in fact no result so far, therefore we did not include them into tables. In the meantime, one test ruptured and its result is newly included in table 3. And, when we compare the results with other authors, there are hardly much more creep results.

8. Table 4: the similar problem. why only 3 samples at 650?

Yes, we tested only one test specimen for each condition. Mainly it was due to the lack of material. But in general, testing matrices changed in the last decades and we test usually only one test piece even for the commercial customers.

9. Figure 8 should be merged with figure 1.

Yes, I did it.

10.  Microstructural analysis was good. However, in figure 13 a very unusual scale of 13 micro meter was used. Please double check this

I changed it.

Reviewer 3 Report

Please read comments of the attached file

Author Response

Dear Reviewer 3,

thank you for your comments. 

I marked with arrows some of the defects.

I define the austenite in the figure foot.

I consider, Hip could be a good option to reduce unmelted powder and some pores, not the one at the surface. But my opinion is based on information from the literature, we didn´t use it in our research. 

I changed SLM to LB-PBF according to new standard, thank you for this notice. I updated the work about your notice

We added new information to conclusions.

Low plasticity is then due to numerous defects and imperfections like pores and unmelted powder, I think it is a fact. Dr. Coro in their interesting work found method how it is possible to evaluate them. I think, possibility to degrease the deffects is in changing parameters and technology of printing.

Sincerely

Šárka Hermanová

Reviewer 4 Report

Manuscript numbered “metals-1820316” has been reviewed:

The introduction needs some improvements.

Please add a suitable scale bar for figures.

Please use the standard terminologies of ASTM/ISO 52900 for the correct terminologies of the Additive Manufacturing process. Use Laser-based powder bed fusion (LB-PBF) instead of selective laser melting (SLM).

Results have been just reported, please compare your finding with other research.

Please add some information about the fabrication process parameter and powder size.

Please add more information about the fracture mechanism.

Compare the results with other 3D Printing methods.

Fallowing papers are suggested for the introduction and result section:

Effects of fused filament fabrication parameters on the manufacturing of 316L stainless-steel components: geometric and mechanical properties

Evolution of temperature and residual stress behavior in selective laser melting of 316L stainless steel across a cooling channel

Author Response

Dear Reviewer 4,

The introduction needs some improvements. 

Yes, i did it.

Please add a suitable scale bar for figures.

I added them.

Please use the standard terminologies of ASTM/ISO 52900 for the correct terminologies of the Additive Manufacturing process. Use Laser-based powder bed fusion (LB-PBF) instead of selective laser melting (SLM).

I changed in the revised manuscript. Now there is LB-PBF.

Results have been just reported, please compare your finding with other research.

Yes, I did it in the revision of the manuspript. 

Please add some information about the fabrication process parameter and powder size.

I added them.

Please add more information about the fracture mechanism.

We added informations about creep behavior and fracture mechanism.

Compare the results with other 3D Printing methods.

Some results of our work is compared now, creep and tensile test.

Fallowing papers are suggested for the introduction and result section:

Effects of fused filament fabrication parameters on the manufacturing of 316L stainless-steel components: geometric and mechanical properties

Evolution of temperature and residual stress behavior in selective laser melting of 316L stainless steel across a cooling channel.

Thank you for new idea.   

Sincerely

Šárka Hermanová et. all

Round 2

Reviewer 1 Report

In general, my comments have been corrected. It would be possible to work more on the Introduction section. But in general, the article has become more understandable and can be published in my opinion.

Author Response

Dear reviewer 1,

thank you for your minority comments. I changed only 2.3.4 Stress rupture test method according to comments of Reviewer 2.

Kind regards,

Šárka Hermanová

Reviewer 2 Report

It is not clear point number 7. Why four samples for one parameter and why six samples for another parameter

Author Response

Dear Reviewer 2,

Thank you for your comments. I changed the 2.3.4. Stress rupture test method, where I describe how we tested. We tested six stress values, for two temperature and both materials. Some of the tests are still running and these are not included in results and tables. I hope that this answer and the change in 2.3.4 is sufficient understandable.

Kind regards,

Šárka Hermanová

Reviewer 3 Report

paper was improved

Author Response

Dear reviewer 3,

 thank you for your approval of revision 1. I prepared revision 2 with changes in 2.3.4. Stress rupture test method according to comments Reviewer 2. 

Kind regards,

Šárka Hermanová

Reviewer 4 Report

The authors have addressed the issues raised previously, and the manuscript is suitable for publication in its current form.

Author Response

Dear reviewer 4,

 thank you for your answer that manuscript is suitable for publication in its current form. But I prepared revision 2 with changes in 2.3.4. Stress rupture test method according to comments Reviewer 2. 

Kind regards,

Šárka Hermanová